# Link prediction accuracy on real-world networks under non-uniform missing-edge patterns

Xie He[1]*, Amir Ghasemian[2], Eun Lee[3], Alice C. Schwarze[1], Aaron Clauset[4,5], Peter J. Mucha[1]*

**1** Department of Mathematics, Dartmouth College, Hanover, NH, United States of America, **2** Yale Institute for Network Science, Yale University, New Haven, CT, United States of America, **3** Department of Scientific Computing at Pukyong National University, Busan, Korea, **4** Department of Computer Science and the BioFrontiers Institute at the University of Colorado, Boulder, CO, United States of America, **5** Santa Fe Institute, Santa Fe, NM, United States of America

* shae.xiehe@gmail.com (XH); peter.j.mucha@dartmouth.edu (PJM)

**Data Availability Statement:** The real world network data used in this study are publicly available on the ICON website and the code for

## Abstract

Real-world network datasets are typically obtained in ways that fail to capture all edges. The patterns of missing data are often non-uniform as they reflect biases and other shortcomings of different data collection methods. Nevertheless, uniform missing data is a common assumption made when no additional information is available about the underlying missing-edge pattern, and link prediction methods are frequently tested against uniformly missing edges. To investigate the impact of different missing-edge patterns on link prediction accuracy, we employ 9 link prediction algorithms from 4 different families to analyze 20 different missing-edge patterns that we categorize into 5 groups. Our comparative simulation study, spanning 250 real-world network datasets from 6 different domains, provides a detailed picture of the significant variations in the performance of different link prediction algorithms in these different settings. With this study, we aim to provide a guide for future researchers to help them select a link prediction algorithm that is well suited to their sampled network data, considering the data collection process and application domain.

## Introduction

Link prediction serves as a valuable tool to correct for missingness inherent in most methods of network data collection. Indeed, many network datasets are only partial samples of the full set of connections present in the real world and, as such, link prediction is a prevalent technique in network analysis across various domains [1, 2], encompassing studies in social networks [3], biological networks [4, 5], information networks [6], and epidemic networks [7]. A multitude of methods have been developed for link prediction over the years, including local similarity indices [8, 9], network embedding techniques [10, 11], matrix completion approaches [12, 13], ensemble learning methods [14], and other methodologies [15, 16].

In many studies, researchers assume that edges are lost uniformly at random; that is, the resulting missing-edge pattern is uniform. Indeed, it is especially common for researchers to

implementing the method and reproducing the numerical experiments presented here can be found online at https://github.com/hexie1995/NonUnifromSampling. Please refer to commit number 4ee24f0 on May 08, 2024, with DOI: 10.5281/zenodo.11661282.

**Funding:** This work was supported in part by the Army Research Office under MURI award W911NF-18-1-0244 (X.H., A.C.S. and P.J.M.), the National Institutes of Health under award R01TW011493 (X.H. and P.J.M.), the National Science Foundation under Grant Nos. 2308460 (X.H. and P.J.M.), and 2030859 to the Computing Research Association for the CIFellows Project (A.G.), and the National Research Foundation of Korea (NRF) grant funded by the Korea government (MSIT) (No.RS-2022-00165916) (E.L.), and Global - Learning & Academic research institution for Master's·PhD students, and Postdocs(LAMP) Program of the National Research Foundation of Korea(NRF) grant funded by the Ministry of Education(No. RS-2023-00301702) (E.L.). The funders had no role in study design, data collection and analysis, decision to publish, or preparation of the manuscript.

**Competing interests:** The authors have declared that no competing interests exist.

assume a uniform missing-edge pattern [17] when no information about the true missing-edge pattern is available. However, missing-edge patterns in real-world settings are likely to be non-uniform due to the nature of the underlying system or the sampling process used to gather the data [18]. In cases where one has knowledge about the missing-edge pattern, e.g. when one knows the sampling method used in the data collection process, we expect that a researcher might achieve better link prediction performance by selecting a suitable link prediction method for the expected missing-edge pattern in their dataset. For example, missing data in social network analysis can result from survey non-response among a closely connected group of individuals, rather than being uniformly distributed [19], or through egocentric sampling [20]. Moreover, missing friendship relationships often occur among outliers within a group who are not closely connected to the rest of the group, unlike other nodes [21]. In protein-protein interaction networks, the absence of a few proteins can lead to missing edges related to all of the missing proteins, therefore providing no information about the relevant nodes [22]. Assuming uniform random missing-edge patterns can thus be inappropriate for link prediction in real-world networks.

Some efforts have been made to address the issue of non-uniform missing-edge patterns in link prediction; in particular, a recent study proposed a computationally efficient link prediction algorithm for egocentrically sampled networks [20]. However, systematic comparisons between missing-edge patterns and their effects on link prediction accuracies are lacking.

We aim here to address this shortcoming through a thorough study over numerous networks, missing-edge patterns, and link prediction algorithms, with a focus on whether a selected prediction method is appropriate for a given missing-edge pattern and whether any method consistently achieves good results across different missing-edge patterns and domains.

To address these questions, we conducted a systematic study with 250 structurally diverse real-world network datasets from ICON [23]. These networks originate from six different disciplinary domains: i) biological (47%), ii) economic (4%), iii) informational (5%), iv) social (30%), v) technological (10%), and vi) transportation (4%). To simulate a diverse range of missing-edge patterns across various real-world scenarios, we applied 20 distinct missing-edge patterns grouped into five different categories: edge-based, node-based, depth-first search (DFS), neighbor-based, and jump-based methods. We then assessed the effectiveness of 9 link prediction algorithms, encompassing 4 different families: local similarity measures, matrix factorization, embedding methods, and ensemble learning.

To the best of our knowledge, our study is the first attempt to systematically compare the effects of different domains and missing-edge patterns on the accuracy of link prediction across several hundred real-world networks. Our findings emphasize the importance of taking into account the dataset domain and associated missing-edge pattern, particularly how the data was sampled, when choosing an appropriate prediction algorithm in a specific setting. We conclude by discussing the limitations of our study and identifying opportunities for further improvement beyond these results.

## Materials and methods

### General pipeline

We use the area under the receiver operating characteristic curve (AUC), a standard in this field [8], as our primary measure for evaluating link prediction performance. The AUC scores provide a context-agnostic measure of method robustness, capturing the ability to distinguish between a missing edge (a true positive) and a non-edge (a true negative) [24], while allowing for easy comparison with the existing link prediction literature. Although other accuracy measures can offer insights into a predictor's performance in specific scenarios (e.g., precision and

recall at certain thresholds), we leave such investigation for future work. Unless otherwise stated, the reported AUC scores in our results are averaged over 5 randomized runs on each network using training, validation, and testing sets constituting roughly 64%, 16%, and 20%, respectively, of the original network data, sampled as follows. While conducting a larger number of independent trials would be ideal, we note that running the present suite of numerical experiments, sampling 250 datasets across 20 different missingness patterns and employing 9 link prediction algorithms, requires approximately $10^5$ CPU core hours. Emphasizing that our reported results are at the disciplinary domain level, not the individual network level, our calculations yield 50 independent AUC scores for each of the smallest domains (economic and transportation).

Our experimental pipeline begins with a given simple graph $G = (V, E)$ consisting of a set $V$ of $n$ nodes and a set $E$ of $m$ edges. For each run of our experiment on $G$, we first draw 20% of the edges uniformly at random from $E$ to define the test set $Y$, which will be the same for each missingness pattern and link prediction algorithm. A major challenge in link prediction using supervised methods is the lack of negative examples (i.e., true non-edges). Following the approach of Ghasemian et al. [25], we consider all non-edges in $G$, i.e., $\tilde{E} = V \times V - E$, as true non-edges. We then sample a testing subset $\tilde{Y}$ of true non-edges from $\tilde{E}$, which, similar to $Y$, remains the same for each missingness pattern and link prediction algorithm.

Next, we define $E' = E - Y$ as the available edges of $G$. We extract the largest connected component from this set, denoted as $LCC(E')$, to ensure fair comparison between the missingness functions we consider, because some of them only apply to connected networks. For each of the 20 missingness patterns we consider in our study, we sample a subset of edges $E_{\text{sample}} \subset LCC(E')$—that is, $E_{\text{sample}}$ is different for each of the missing-edge patterns—targeting the size of $E_{\text{sample}}$ to include $\approx$64% of the edges in the original graph $G$. (We note that some of the missingness functions we use to obtain our samples do not provide sufficient control to obtain precisely this count of edges; in such cases, we pick a sample close to the target size.)

Our task is then to predict, using the sampled edges $E_{\text{sample}}$, which pairs of nodes not connected by the available edges, $X = V \times V - E'$, are actually missing edges (cf. true non-edges; specifically, we test on the node pairs in $Y$ and $\tilde{Y}$). Each link prediction algorithm provides a score function over node pairs $(i, j) \in X$, where a higher score indicates the algorithm asserts a higher likelihood of the node pair being a true missing edge [3]. A subset of the link prediction methods studied here are supervised methods, requiring separate training, validation, and testing sets. In such cases, we let $E_{\text{sample}}$ be the training set and $Z = E' - E_{\text{sample}}$ be the validation set of edges, with the training and validation sets of non-edges both taken from $\tilde{E}' = V \times V - E - \tilde{Y} = \tilde{E} - \tilde{Y}$, while continuing to use $Y$ and $\tilde{Y}$ as our test set. For these supervised methods specifically, we perform 5-fold cross-validation, separately dividing up both the training set and validation set, during the training process. (See also the special note about the use of these sets for the **Top-Stacking** method as described below in **Ensemble Learning**.)

Having defined sets of true positives (edges) and negatives (non-edges), we conduct additional resampling to achieve balanced classes (i.e., edge presence/absence) during both training and testing stages, following the approach outlined by Estabrooks et al. [26] for supervised methods. Specifically, we sample 10,000 edges uniformly at random with replacement to form the positive class, and an equal number of non-edges to form the negative class, in each of our train, validation, and test sets. We additionally emphasize that we perform this sampling of non-edges for training and validation in a manner ensuring that they have empty intersection.

Note, very importantly, these designs described above were crafted with specific consideration for the present study's objectives, including to try to make equitable comparisons

between the results obtained under different missingness functions and different link prediction algorithms. That said, we acknowledge that many other viable choices are possible, and we emphasize that other choices should be considered in settings where the objectives of a study are different.

## Missing-edge patterns

To examine the impact of different missing-edge patterns, we apply 20 different sampling methods from **Little Ball of Fur** [27], a Python library for network sampling techniques that uses a single streamlined framework of sampling functions. We group these 20 methods into 5 different categories: i) edge-based, ii) node-based, iii) depth first search (DFS), iv) neighbor-based, and v) jump-based missingness patterns. Edge-based missingness includes traditional uniform sampling methods and other techniques that primarily focus on edges. These sampling methods correspond to common ideas found in the current literature regarding missing-edge patterns. Node-based missing-edge patterns direct attention toward individual node properties, such as degree centrality and PageRank. These missing-edge patterns can be relevant to real-world scenarios where the sampling procedure is biased towards individuals with either greater or lesser influence. DFS is singled out here due to its distinctive nature in exploring neighborhoods, making it a suitable mimic for scenarios where nodes are sampled along long chains in the network, such as in some observations of criminal activities. Neighbor-based missingness patterns prioritize exploration based on the neighborhood of a set of initial seed nodes. They offer an idealized sampling model for sociological surveys, disease contact tracing, and similar social-network studies. Jump-based missingness patterns introduce a unique element by allowing jumps from one node to another. These sampling methods can thus yield fragmented networks, and they are relevant in real-world scenarios when, for example, including random encounters in the data collection leads to a fragmented social network. These categories aim to capture some of the diverse aspects of missing-data patterns, thereby enhancing the understanding of algorithmic performance across different sampling scenarios.

**Edge-based missingness patterns.** The **Random Edge Sampler** from Krishnamurthy et al. [28] samples edges uniformly at random. (We emphasize again that this method is the most commonly used method in the literature for assessing link prediction methods.) The **Random Node-Edge Sampler** from Krishnamurthy et al. [28] first samples nodes uniformly at random; then for each sampled node, one of its edges is sampled (again uniformly). The **Hybrid Node-Edge Sampler** from Krishnamurthy et al. [28] combines and alternates between the above two methods, sampling some portion of the edges uniformly at random and another portion from the edges incident to a node within a node set that is sampled uniformly at random. The **Random Edge Sampler with Induction** from Ahmed et al. [29] includes two steps: One first samples edges uniformly at random with a fixed probability. The resulting graph has a node set $V'$. One then samples additional edges from the induced subgraph of $G$ on $V'$ uniformly at random until the desired number of edges are obtained. For the fixed probability in the first step, we use a default value of 0.5. For the second step, we construct the induced subgraph following algorithm 1 in [29].

**Node-based missingness patterns.** For each of the following methods, only the edges between nodes that are both sampled will be included. That is, the sampled edges $E_s$ are those that appear in the induced subgraph of the set of sampled nodes. The **Degree Based Node Sampler** from Adamic et al. [30] samples nodes with probability proportional to their degrees. The **Random Node Sampler** from Stumpf et al. [31] samples nodes uniformly at random. Similarly, the **PageRank Based Node Sampler** from Leskovec et al. [32] samples proportional to the PageRank scores of nodes.

**Depth first search missingness pattern.** The **Randomized Depth First Search (DFS) Sampler** [33] starts from a randomly chosen node and adds neighbors to the last-in-first-out queue after shuffling them randomly.

**Neighbor-based missingness patterns.** The following sampling methods include all nodes explored by a walker and all the corresponding edges in the induced subgraph between these sampled nodes, resulting in connected samples. The **Diffusion Sampler** [34] applies a simple diffusion process on the network to sample an induced subgraph incrementally. The **Forest Fire Sampler** [35] is a stochastic snowball sampling method where the expansion is proportional to the burning probability, with a default probability of 0.4. The **Non-Backtracking Random Walk Sampler** [36] samples nodes with a random walk in which the walker cannot backtrack. The **Random Walk Sampler** [37] samples nodes with a simple random walker. The **Random Walk With Restart Sampler** [32] uses a discrete random walk on nodes, with occasional teleportation back to the starting node with a fixed probability. The **Metropolis–Hastings Random Walk Sampler** [38] uses a random walker with probabilistic acceptance condition for adding new nodes to the sampled set, which can be parameterized by the rejection constraint exponent. The **Circulated Neighbors Random Walk Sampler** [39] simulates a random walker, and after sampling all nodes in the vicinity of a node, the vertices are randomly reshuffled to help the walker escape closely-knit communities. The **Randomized Breadth First Search (BFS) Sampler** [33] performs node sampling using breadth-first search: starting from a randomly chosen node, neighbors are added to the queue after shuffling them randomly.

**Jump-based missing-edge patterns.** The **Random Walk With Jump Sampler** [40] is done through random walks with occasional teleporting jumps. The **Random Node-Neighbor Sampler** [32] samples nodes uniformly at random and then includes all neighboring nodes and the edges connecting them as the induced subgraph. The **Shortest Path Sampler** [41] samples pairs of nodes and chooses a random shortest path between them, including the vertices and edges along the selected shortest path in the induced subgraph. These jump-based missing-edge patterns methods can result in a fragmented sample graph. The **Loop-Erased Random Walk Sampler** [42] samples a fixed number of nodes, and then includes only edges that connect previously unconnected nodes to the sampled set, resulting in an undirected tree.

## Link prediction methods

We employ 9 different link prediction methods. These link predictions are drawn from 4 popular families of algorithms: local similarity indices [8, 9], network embedding [10, 11], matrix completion [12, 13], and ensemble learning [14].

**Network embedding.** The **Node2Vec Dot Product** approach uses a skip-gram-based method to learn node embeddings from random walks in the graph. The dot product between the embeddings of two nodes is used to represent the corresponding edge, which is then used for training [43]. The **Node2Vec Edge Embedding** method also learns node embeddings using skip-gram-based techniques; however, it additionally incorporates bootstrapped edge embeddings and logistic regression to represent the edges, enhancing the training process [43]. The **Spectral Clustering** approach uses spectral embeddings to create node representations from an adjacency matrix [44, 45] and then makes predictions with the embedded feature vector.

**Local similarity indices.** One such method relies on the **Adamic-Adar** index, a local structured measure, for link prediction [46]. Another approach utilizes **Preferential Attachment**, another local structured measure, for link prediction [46]. The **Jaccard Coefficient** can be similarly employed for link prediction [46].

**Ensemble learning.**   The **Top-Stacking** method combines topological features of node pairs and trains a random-forest model for link prediction [25]. It's crucial to highlight that for Top-Stacking, unlike other supervised methods, which directly utilize the validation set for validation and parameter tuning, the training edges for this specific method are instead drawn from the validation set. We take this approach to align with the methodology described in Ghasemian et al. [25], who generate network features with (in the present notation) $E_{\mathrm{sample}}$ for training the classifier on edges selected from $E' - E_{\mathrm{sample}}$. This is consistent with using the validation set for hyperparameter tuning of other supervised methods, insofar as all of the ensemble learning here is being tuned by training on this set. In this manner, Ghasemian et al. aim to mimic the real-world setting, where network features are calculated with the observed network to predict missing edges. This is then similar to testing on the withheld 20%, having computed network features from the available training data.

Indeed, we observe a sometimes drastic over-fitting if we train on edges from the same set used to generate training features; such behavior is perhaps not surprising a posteriori, since building a classifier to detect edges that were included in computing the network features is a different problem from predicting missing edges that were not available to be included in the network feature calculation. We thus emphasize that our calculation of network features from $E_{\mathrm{sample}}$ to train the classifier on edges from $E' - E_{\mathrm{sample}}$ maintains the pipeline as described in Ghasemian et al. [25].

**Matrix completion.**   The **Modularity** method for link prediction starts from Newman and Girvan's modularity maximization for community detection [47]. We then use the obtained communities to measure the empirical densities between and within communities and then predict the most likely missing edges from these densities, similar to [25, 48]. The **MDL-DCSBM** method uses a minimum description length, degree-corrected stochastic block model [49]. As with the Modularity method, one then uses the obtained block structure to predict the most likely missing edges in the manner used in [25, 50].

## Results

### Data

We use publicly available data listed on [51] and provided in clean form in the Github repository for Ghasemian et al. [25]. To reduce the computational cost of our simulation study, we consider a subset of the data that includes 119 biological networks, 9 economic networks, 12 informational networks, 74 social networks, 26 technological networks, and 10 transportation networks. This subset of 250 networks, which includes networks with more than 20 but less than 900 nodes, has a mean of 492 nodes and 1110 edges, with a mean node degree of 5.28. This diversity of network data, encompassing a wide array of domains and collected through various methodologies, provides a robust foundation for evaluating sampling methods and link prediction algorithms. Assessing performance across this extensive dataset spectrum allows us to investigate how missing-edge patterns and link prediction relate to one another in different domains as opposed to considering only a single domain. All the relevant code can be found at https://github.com/hexie1995/NonUnifromSampling.

### Domain and algorithm variability

For a comprehensive assessment of the performance and variability of various link prediction algorithms across missing-edge patterns and diverse domains, in Fig 1 we visualize AUC scores obtained under 5 repeats across 120 distinct combinations of domains (6) and sampling methods (20). The average AUC scores under these 120 settings are listed in the S1-S6 Tables in S1 File. Our results show that the Top-Stacking method achieves the best performance in

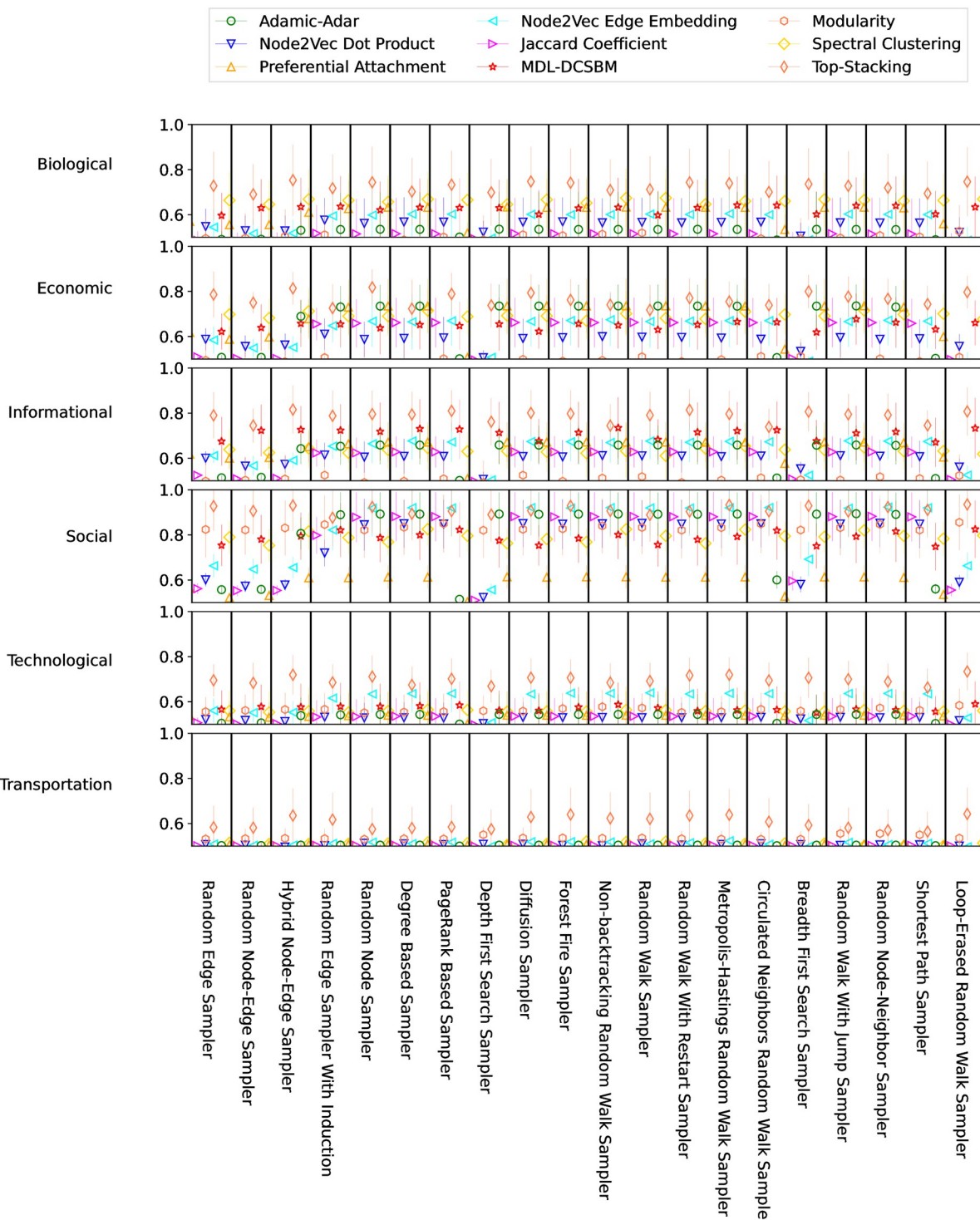

**Fig 1. AUC scores over 5 runs on each network for 9 link prediction algorithms on samples obtained by 20 methods.** The 250 different networks are grouped into 6 domains (arranged vertically). Symbols indicate mean AUCs, with standard deviations shown by vertical bars. The sampling methods are listed along the bottom of the figure. The prediction methods are marked with different colors, as indicated in the legend at the top.

109 instances. The Node2Vec Edge Embedding method performs best in 9 instances. Preferential Attachment is the best-performing method in 2 instances, with Adamic-Adar equaling its performance in 1 of those 2 instances.

The results visually depicted in Fig 1 demonstrate that, within a specific dataset domain, a particular or a group of link prediction method tends to outperform others (see also S1-S6 Tables in S1 File). For example, MDL-DCSBM and Top-Stacking tend to have consistently higher AUC scores than the other prediction algorithms for informational networks.

For social networks, importantly, the AUC scores across missing-edge patterns exhibit a clear and distinct pattern for different missing-edge patterns (see next section for detailed discussion). Nevertheless, almost all considered link prediction algorithms yield good prediction results for a majority of missing-edge patterns, while Preferential Attachment has the smallest predictive power among the considered methods. This observation is consistent with the findings of other researchers who suggested that structural properties that are commonly associated with social networks, such as high clustering coefficients and heavy-tailed degree distributions, positively affect the accuracy of link prediction [52–54]. These results provide supporting evidence for a previous observation made by several papers, including Ghasemian et al. [25] and Menand et al. [54], that most link prediction methods can achieve high AUC scores on social networks, and suggest that this claim is also true for various missing-edge patterns (excluding some). We thus stress that further comparisons of different link prediction algorithms should continue to include many types of real-world networks, rather than social networks only.

In economic networks, Top-Stacking stands out as the best-performing method overall, while Preferential Attachment consistently secures the second place, demonstrating AUC scores close to and sometimes better than Top-Stacking. It is noteworthy that Preferential Attachment has a specific efficacy in economic networks, demonstrating good predictive power despite being a simple topological predictor and a relatively weak predictor in other domains (even social networks).

Across biological networks, which comprise the majority of the networks under consideration (119 out of 250), informational networks, technological networks, and transportation networks, Top-Stacking consistently produces superior AUC scores compared to all other methods. In biological networks, Spectral Clustering emerges as the second-best performer, while MDL-DCSBM is second best in informational networks and Node2Vec Edge Embedding is second best in technological networks.

We note that the link prediction accuracies are typically notably less accurate for technological and transportation networks across sampling methods and link prediction algorithms compared to accuracies for networks from the other four disciplinary domains. We suspect two possible reasons for this. First, the limited number of these types of networks in our dataset collection might bias the results, though in hypothesizing such an effect we note that the numbers of networks we study from the economic and informational domains are also small. Perhaps more likely, there is some systematic network feature or properties about data from these domains that could be found to be related to lower link prediction accuracy. As one for example, we note that performing link prediction in very sparse networks can lead to additional challenges (see, e.g., [55, 56]).

Summarizing our findings across these six domains, we observe under most circumstances that Top-Stacking consistently provides the most accurate link predictions or, at the very least, achieves comparable performance to the leading method in each domain.

## Influence of missingness patterns

We now shift our focus towards the variations of AUC scores across different dataset domains under various missing-edge patterns. To provide a comprehensive visualization, we show the performance of link prediction methods for the five different missingness categories (Fig 2). We collect the AUC scores as box plots from their respective categories to better discern and illustrate the differences across various missing-edge patterns. A complete array of box plots for all missingness patterns is presented in the S1-S5 Figs in S1 File.

The first notable observation is the distinct behavior exhibited by the DFS-based missing-edge pattern. For DFS-sampled networks, regardless of the choice of link prediction algorithm or network domain, we consistently observe slightly lower AUC scores than for other sampling techniques. In terms of link prediction methods, generally speaking, Modularity, MDL-DCSBM, Spectral Clustering, and Top-Stacking are less impacted by the DFS based missingness pattern, displaying smaller decreases in performance than the other methods. Nevertheless, all these methods still display a decay in performance and smaller variance. Concerning network domains, the only domain that doesn't exhibit a significant decline in performance for DFS-based missingness is the transportation networks. However, the performance for transportation networks is not initially robust, making it challenging to utilize this as a comparison metric.

To further explore the variation behind the sampling methods and their effect on the link prediction algorithms for datasets across various domains, we perform Principal Component Analysis (PCA) using the AUC scores as features to characterize each of the 20 sampling methods, with PCA performed separately for each of the 6 data domains and 9 link prediction algorithms Fig 3. That is, for each sampling method and prediction algorithm, we use the AUC scores obtained (averaged over 5 runs) from the different networks in that domain as the feature vector. We then perform PCA on the set of feature vectors across different sampling methods, performed separately for each prediction algorithm in each data domain. For example, the top left corner panel visualizes the first two principal component scores of the 20 sampling methods obtained from the AUCs of Adamic-Adar on the 119 biological networks—i.e., the feature vector of a sampling method includes the 119 AUC scores stacked together.

The results further emphasize DFS as an outlier among the considered missing-edge patterns, with DFS standing out in most of the single-panel PCA plots. We hypothesize that this is due to inherent structural features of DFS samples [33]. Based on our findings, we suggest that DFS samples are a particularly interesting and important topic to study. The missing-edge patterns in real-world network data could very well be focused on the missing information around one particular egocentric subgraph or along a particular line of interactions, such as criminal and terrorist activities [57]. For some real-world cases, DFS may be much more suitable to mimic the real missing-edge pattern in the data. Such cases can motivate the study of the stability and robustness of link prediction algorithms under such biased samples of edges.

Another notable observation is the significant impact of missing-edge patterns on the performance of link prediction algorithms in social networks. We observe a noticeable increase in performance when transitioning from edge-based missingness patterns to node-based missingness patterns during the sampling of social networks. Furthermore, although the average results for node-based, neighbor-based, and jump-based missingness patterns are close to each other in social networks, there is a clear difference in variance. Jump-based patterns exhibit much larger variance than both neighbor-based and node-based methods, with the node-based method showing the smallest variance. This suggests that when modeling missingness patterns in social networks, node-based sampling could be a more realistic choice than other missing-edge patterns.

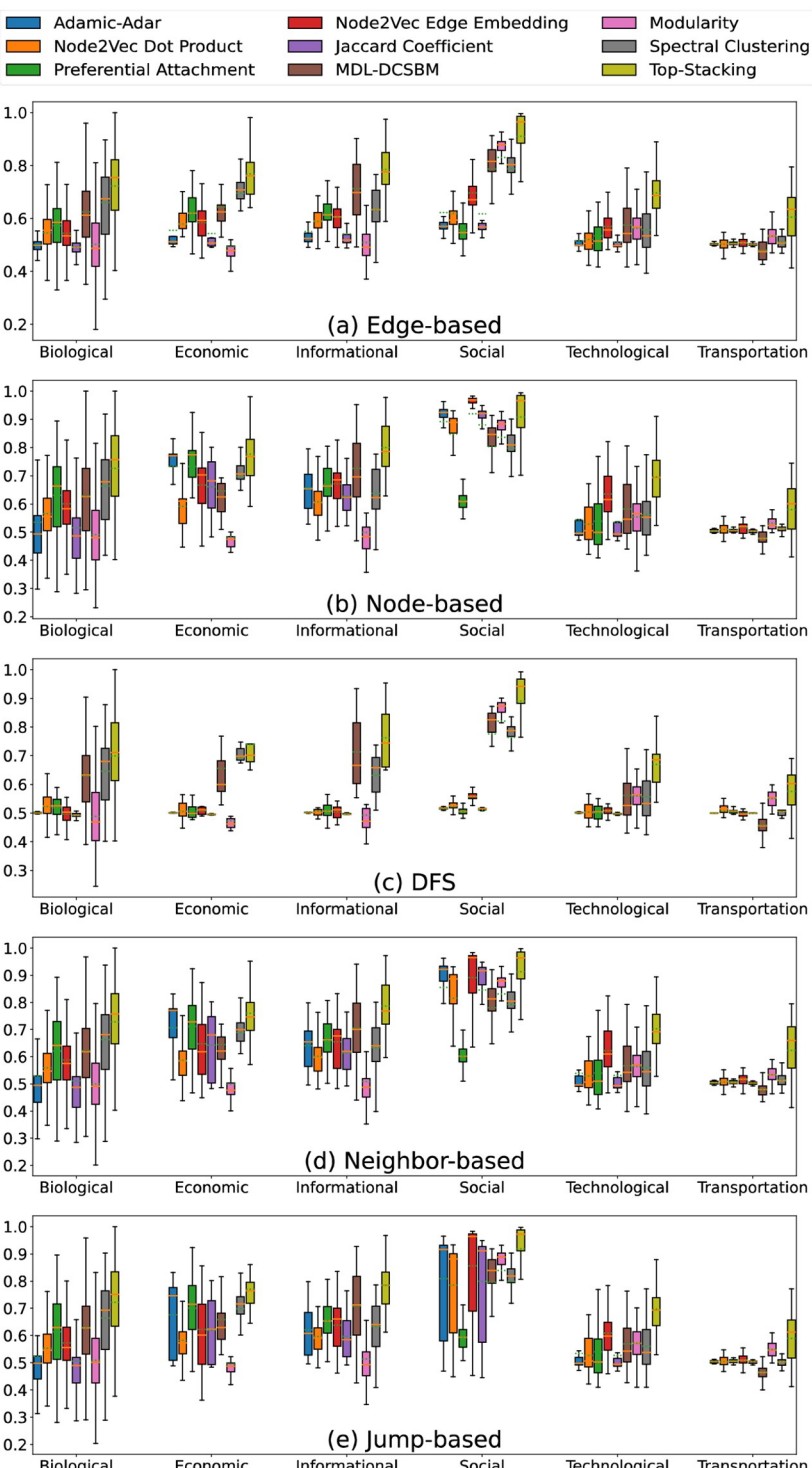

**Fig 2. Box plots of AUCs from different link prediction methods for different families of missingness patterns, grouped by network domain.** The plotted whiskers indicate the extreme range of values, up to the constraint of being no longer than 1.5x the interquartile range; any outliers beyond this range have been removed for improved visibility. Corresponding results for individual missingness patterns are plotted in the same manner in S1-S5 Figs in S1 File.

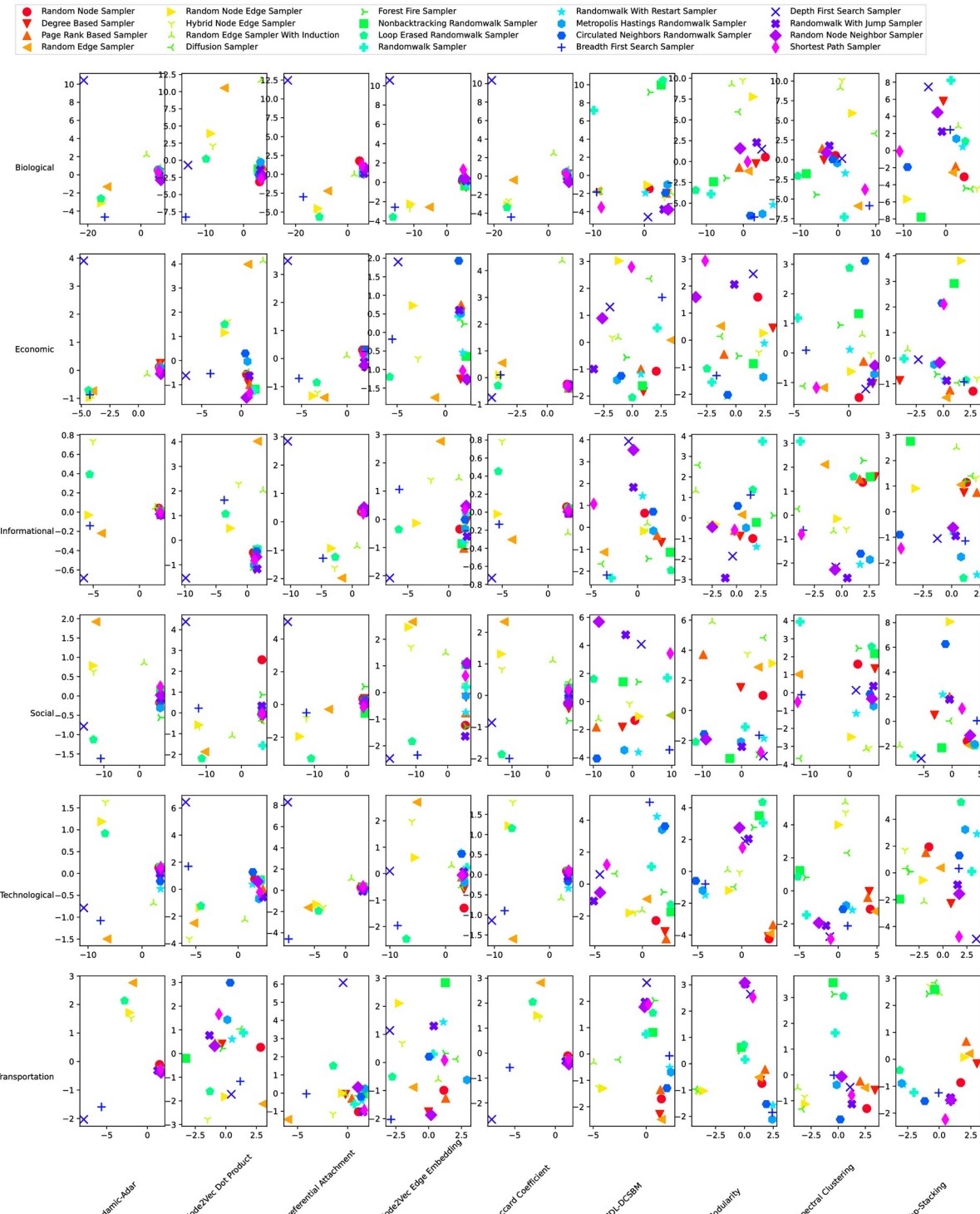

**Fig 3. PCA scores (PC1 horizontal, PC2 vertical) of different sampling methods under different prediction algorithms and different dataset domains.** Each panel considers a single link prediction method within a single dataset domain, taking as features the full set of AUC scores (averaged over 5 runs) of that prediction method across the networks in that domain for each sampling method, marked with different colors and symbols as indicated in the legend.

For networks from other domains, the distribution of AUC scores is distinctly different for missingness patterns as well. For instance, in economic networks, we observe again that for both Adamic-Adar and Preferential Attachment, node-based missingness patterns lead to better AUC scores than edge-based missingness patterns. Furthermore, the variance for all methods increases when transitioning from edge-based methods to node-based, neighbor-based, or jump-based methods, with jump-based methods again exhibiting the largest variance. The discrepancy in variance becomes even more obvious when examining the differences between individual missingness patterns rather than considering them collectively (see S1-S5 Figs in S1 File).

The apparent differences in variance underscore that it is important to consider missing-edge patterns when selecting appropriate link prediction algorithms. This observation aligns with the insights gleaned from S1-S6 Tables in S1 File. When the missing-edge pattern is uncertain or unknown, one can generally consider link prediction via Top-Stacking as a prudent starting point, given its consistent performance across different missing-edge patterns. One can then further use knowledge about the domain of the considered dataset(s) to refine one's choice of a link prediction method.

For example, in the context of social networks, Node2Vec Edge Embedding yields excellent performance when dealing with node-based missingness patterns. However, its performance significantly deteriorates when confronted with edge-based missingness patterns. Consequently, deploying both Node2Vec Edge Embedding and Top-Stacking as the primary link prediction methods ensures the utilization of the most effective approach. This selection approach eliminates the need for exhaustive testing across multiple algorithms in cases where the domain of the dataset is known but the missingness patterns is unknown.

Finally, when information about the missing-edge pattern is also available, this information can be important for selecting an appropriate link prediction method for a given data set, and for assessing the level of confidence one might have in the link-prediction results. For instance, we again highlight the overall decreased link prediction accuracies encounted with DFS samples across most domains and algorithms. As another example visible in Fig 2, we note the relatively good performance of the Adamic-Adar predictor on economic networks for some pattern families (e.g., node-based and neighbor-based). That is, both the domain of a network and the missingness pattern encountered can have a substantial impact on link prediction performance.

## Discussion

Our study establishes key principles for selecting link prediction algorithms in scenarios where the missing edge pattern is pre-determined or known. Specifically, our results also demonstrate that (1) it is crucial to consider the domain of a data set of interest when selecting an appropriate link prediction method and (2) knowledge about missing-edge patterns can anchor a researcher's confidence in their prediction outcome.

Across domains, we consistently observe important performance discrepancies and large variances depending on the chosen missingness patterns. This emphasizes the need for caution during the sampling process when reducing the size of a network or studying the performance of link prediction algorithms. Understanding how edges are missing or sampled can be crucial in determining the most suitable link prediction algorithms to employ. Additionally, this consideration influences the design of experiments in studying missingness patterns. Researchers should carefully assess whether uniform random sampling is appropriate for the scope of their studies, instead of resorting to it automatically.

Of course, real-world data are typically not obtained by sampling a complete data set explicitly via one of our considered samplers. Instead, it is common that the missing-edge pattern is unknown. While no single method excels across all domains and missing-edge patterns, our results suggest that Top-Stacking is the most robust general predictor among the considered datasets when the missing-edge pattern is unknown. It produces the best results in over 90% of our 120 cases (across 6 domains and 20 different missingness patterns), and good (but not the best) results in many other cases. When the network domain and missing-edge pattern are known, more specialized predictors may produce better results, such as Preferential Attachment in the case of economic networks when the missingness pattern is based on node degrees. Another similar example is that for social networks, Node2Vec Edge Embedding appears to surpass Top-Stacking under some settings, depending on the missing edge pattern.

We limit our study to the sampling methods in **Little Ball of Fur** [27], and it is important to acknowledge that these sampling methods serve as models rather than exact replicas of missing-edge patterns in real-world data. We have categorized these sampling methods into different groups that aim to mimic genuine data loss scenarios, but conclusions made based on these models should consequently be approached with caution, since they are neither precisely capturing real-world sampling nor are they exhaustive. For instance, the authors of [58] discussed link prediction for isolated nodes in both static and temporal networks.

Neural network methods have also been successfully used for link prediction to recover missing edges [59]. However, recent publications [25, 60] have raised concerns about their performance relative to more traditional algorithms, as well as their lack of feature explainability, which poses challenges in understanding their performance. While we recognize that neural network methods can deliver outstanding classification performance in many settings, we are unaware of any specific neural network methods for link prediction appropriate to the present task, specifically, without the need for any additional node-level metadata, that exists in publicly available code, and that focuses on feature explainability. Consequently, this study excludes neural network methods and instead prioritizes the understanding of behaviors and features beneficial for diverse real-world applications. Nevertheless, we acknowledge the limitation that, in comparing the overall impact on the accuracy of different link prediction algorithms under different sampling settings and across different domains, an appropriate neural network method might provide a different pattern of variation in these observed accuracies. Conducting a comprehensive examination of neural network methods and comparing them could provide valuable insights for future research.

Another important limitation is that in this study, we solely focused on categorizing the data based on their domain type. It is highly possible that domains serve as proxies for similarities in other measurable network features. Future research endeavors may aim to identify and categorize the network features underlying the diverse behaviors uncovered in this study more clearly. This identification would enable future practitioners to make informed decisions about which link prediction method to employ based on measurements of their sampled network.

Another potentially promising avenue for future research would be a generalization of our study to multilayer networks and temporal networks, given their increasing importance (see, e.g., [61–65] for various methods of link prediction in these settings). For example, it seems particularly important to explore the variability in sampling methods and their impact on prediction accuracy as the amount of temporal information increases, because the missingness pattern might be different for each of the temporal layers. Additionally, we anticipate that it will be beneficial to establish theoretical relationships between link predictability, different sampling settings, and the influence of different network features on prediction outcomes.

## Supporting information

**S1 File. Supporting information that contains S1-S5 Figs and S1-S6 Tables.**
(PDF)

## Author Contributions

**Conceptualization:** Xie He, Amir Ghasemian, Eun Lee, Aaron Clauset, Peter J. Mucha.

**Data curation:** Xie He, Amir Ghasemian.

**Formal analysis:** Xie He.

**Funding acquisition:** Peter J. Mucha.

**Investigation:** Xie He, Amir Ghasemian, Eun Lee, Alice C. Schwarze.

**Methodology:** Xie He, Amir Ghasemian, Eun Lee, Alice C. Schwarze, Aaron Clauset, Peter J. Mucha.

**Software:** Xie He.

**Supervision:** Aaron Clauset, Peter J. Mucha.

**Validation:** Xie He, Amir Ghasemian, Eun Lee, Alice C. Schwarze, Peter J. Mucha.

**Visualization:** Xie He, Amir Ghasemian, Eun Lee, Alice C. Schwarze, Peter J. Mucha.

**Writing – original draft:** Xie He.

**Writing – review & editing:** Amir Ghasemian, Eun Lee, Alice C. Schwarze, Aaron Clauset, Peter J. Mucha.

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
