## [Decision Letter · Decision Letter 0]

4 Jun 2024

PONE-D-24-17386Link Prediction Accuracy on Real-World Networks Under Non-Uniform Missing-Edge PatternsPLOS ONE

Dear Dr. HE,

Thank you for submitting your manuscript to PLOS ONE. After careful consideration, we feel that it has merit but does not fully meet PLOS ONE’s publication criteria as it currently stands. Therefore, we invite you to submit a revised version of the manuscript that addresses the points raised during the review process.

The first reviewer only has some minor suggestions to improve the article. In addition to some technical questions, the second reviewer raised two larger issues: (1) the sampling approach; and (2) missing more recent literature in the graph theory/machine learning area. The first issue may stem from a misunderstanding, but it would be good to clarify this. The second issue might benefit from greater clarity on the motivation for selecting the 9 link prediction algorithms in your study, and for not selecting other link prediction algorithms.

We look forward to receiving your revised manuscript.

Kind regards,

Vincent Antonio Traag, Ph.D.

Academic Editor

PLOS ONE

Journal Requirements:

   "This work was supported in part by the Army Research Office under MURI award W911NF-18-1-0244 (X.H., A.C.S. and P.J.M.), the National Institutes of Health under award R01TW011493 (X.H. and P.J.M.), the National Science Foundation under Grant Nos.\\ 2308460 (X.H. and P.J.M.) and 2030859 to the Computing Research Association for the CIFellows Project (A.G.), and the National Research Foundation of Korea (NRF) grant 445 funded by the Korea government(MSIT) (No. RS-2022-00165916), and Learning \\& Academic research institution for Master’s, PhD students, and Postdocs (LAMP) Program of the National Research Foundation of Korea (NRF) grant funded by the Ministry of Education(No. RS-2023-00301702) (E.L.)."

Reviewers' comments:

Reviewer's Responses to Questions

**Comments to the Author**

1. Is the manuscript technically sound, and do the data support the conclusions?

Reviewer #1: Yes

Reviewer #2: Partly

2. Has the statistical analysis been performed appropriately and rigorously? 

Reviewer #1: Yes

Reviewer #2: No

3. Have the authors made all data underlying the findings in their manuscript fully available?

Reviewer #1: Yes

Reviewer #2: Yes

4. Is the manuscript presented in an intelligible fashion and written in standard English?

Reviewer #1: Yes

Reviewer #2: Yes

5. Review Comments to the Author

Reviewer #1: This manuscript validates link prediction methods with empirical data and a variety of methods to select and hide edges. It is an important reference work for network science and graph learning. The manuscript is clear and well-written. The methods are sound. There is no reason to request a revision, it could be published as it is.

If anything should be polished: 1. The box-and-whiskers plots have too many outliers. Sometimes, the outliers distract the viewer from seeing the patterns shown by the boxes. These could either be plotted without outliers or replaced by other visualization techniques. 2. Trellis plots like Fig. 3 should have the same limits on the axes for readers to compare the panels efficiently.

Reviewer #2: The author indeed aims to extract the applicability of several classic complex network link prediction algorithms from a vast dataset, further elucidating the underlying mechanism of missing edges. However, the article lacks depth in both physical understanding and computational thoroughness. Firstly, the rationale for network sampling is questionable; if it was to save time, this is unlikely to convince readers. While the author attempts to explain, it fails to ensure that the structural features of the sampled network resemble those of the original, instead resorting to various computational strategies, employing numerous sampling techniques to compensate for technical shortcomings. Paradoxically, the more techniques employed, the murkier the underlying principles become, leading to a superficial analysis. Secondly, I note the low predictive accuracy for technological and transportation networks, which I suspect relates to network sparsity. References to 'Shang, K. K., Li, T. C., Small, M., Burton, D., & Wang, Y. (2019). Link prediction for tree-like networks. Chaos: An Interdisciplinary Journal of Nonlinear Science, 29(6); and Shang, K. K., & Small, M. (2022). Link prediction for long-circle-like networks. Physical Review E, 105(2), 024311' might provide insight.

Moreover, since 2019, there has been a surge in utilizing graph theory combined with machine learning and neural networks for link prediction, hence the inclusion of such contemporary algorithms is recommended. Proceeding through the article, I observe a glaring absence of a comprehensive and meticulous differentiation of the mechanisms behind missing links in networks. A tabular summary to address this gap is advisable. Lastly, the choice to independently calculate the Area Under the Curve (AUC) only five times is puzzling, given that since the advent of link prediction techniques, 100 independent calculations have become a standard requirement to ensure statistical robustness.

6. PLOS authors have the option to publish the peer review history of their article (what does this mean?). If published, this will include your full peer review and any attached files.

Reviewer #1: No

Reviewer #2: **Yes: **Ke-ke Shang

---

## [Author Response · Author response to Decision Letter 0]

14 Jun 2024

Please see the Response to Reviewer Letter in the documents for detailed responses.

---

## [Editor Report · Decision Letter 1]

26 Jun 2024

Link Prediction Accuracy on Real-World Networks Under Non-Uniform Missing-Edge Patterns

PONE-D-24-17386R1

Dear Dr. HE,

We’re pleased to inform you that your manuscript has been judged scientifically suitable for publication and will be formally accepted for publication once it meets all outstanding technical requirements.

Kind regards,

Vincent Antonio Traag, Ph.D.

Academic Editor

PLOS ONE

---

## [Editor Report · Acceptance letter]

9 Jul 2024

PONE-D-24-17386R1 

PLOS ONE

Dear Dr. HE, 

I'm pleased to inform you that your manuscript has been deemed suitable for publication in PLOS ONE. Congratulations! Your manuscript is now being handed over to our production team.

Kind regards, 

on behalf of

Dr. Vincent Antonio Traag 

Academic Editor

PLOS ONE